# Plan, Attend, Generate:
# Planning for Sequence-to-Sequence Models

**Francis Dutil**[*]
University of Montreal (MILA)
frdutil@gmail.com

**Caglar Gulcehre**[*]
University of Montreal (MILA)
ca9lar@gmail.com

**Adam Trischler**
Microsoft Research Maluuba
adam.trischler@microsoft.com

**Yoshua Bengio**
University of Montreal (MILA)
yoshua.umontreal@gmail.com

## Abstract

We investigate the integration of a planning mechanism into sequence-to-sequence models using attention. We develop a model which can plan ahead in the future when it computes its alignments between input and output sequences, constructing a matrix of proposed future alignments and a commitment vector that governs whether to follow or recompute the plan. This mechanism is inspired by the recently proposed strategic attentive reader and writer (STRAW) model for Reinforcement Learning. Our proposed model is end-to-end trainable using primarily differentiable operations. We show that it outperforms a strong baseline on character-level translation tasks from WMT'15, the algorithmic task of finding Eulerian circuits of graphs, and question generation from the text. Our analysis demonstrates that the model computes qualitatively intuitive alignments, converges faster than the baselines, and achieves superior performance with fewer parameters.

## 1   Introduction

Several important tasks in the machine learning literature can be cast as sequence-to-sequence problems (Cho et al., 2014b; Sutskever et al., 2014). Machine translation is a prime example of this: a system takes as input a sequence of words or characters in some source language, then generates an output sequence of words or characters in the target language – the translation.

Neural encoder-decoder models (Cho et al., 2014b; Sutskever et al., 2014) have become a standard approach for sequence-to-sequence tasks such as machine translation and speech recognition. Such models generally *encode* the input sequence as a set of vector representations using a recurrent neural network (RNN). A second RNN then *decodes* the output sequence step-by-step, conditioned on the encodings. An important augmentation to this architecture, first described by Bahdanau et al. (2015), is for models to compute a soft alignment between the encoder representations and the decoder state at each time-step, through an *attention* mechanism. The computed alignment conditions the decoder more directly on a relevant subset of the input sequence. Computationally, the attention mechanism is typically a simple learned function of the decoder's internal state, e.g., an MLP.

In this work, we propose to augment the encoder-decoder model with attention by integrating a planning mechanism. Specifically, we develop a model that uses planning to improve the alignment between input and output sequences. It creates an explicit plan of input-output alignments to use at future time-steps, based

---

[*] denotes that both authors (CG and FD) contributed equally and the order is determined randomly.

on its current observation and a summary of its past actions, which it may follow or modify. This enables the model to plan ahead rather than attending to what is relevant primarily at the current generation step. Concretely, we augment the decoder's internal state with (i) an *alignment plan* matrix and (ii) a *commitment plan* vector. The alignment plan matrix is a template of alignments that the model intends to follow at future timesteps, i.e., a sequence of probability distributions over input tokens. The commitment plan vector governs whether to follow the alignment plan at the current step or to recompute it, and thus models discrete decisions. This is reminiscent of macro-actions and options from the hierarchical reinforcement learning literature (Dietterich, 2000). Our planning mechanism is inspired by the *strategic attentive reader and writer* (STRAW) of Vezhnevets et al. (2016), which was originally proposed as a hierarchical reinforcement learning algorithm. In reinforcement-learning parlance, existing sequence-to-sequence models with attention can be said to learn reactive policies; however, a model with a planning mechanism could learn more proactive policies.

Our work is motivated by the intuition that, although many natural sequences are *output* step-by-step because of constraints on the output process, they are not necessarily *conceived* and *ordered* according to only local, step-by-step interactions. Natural language in the form of speech and writing is again a prime example – sentences are not conceived one word at a time. Planning, that is, choosing some goal along with candidate macro-actions to arrive at it, is one way to induce *coherence* in sequential outputs like language. Learning to generate long coherent sequences, or how to form alignments over long input contexts, is difficult for existing models. In the case of neural machine translation (NMT), the performance of encoder-decoder models with attention deteriorates as sequence length increases (Cho et al., 2014a; Sutskever et al., 2014). A planning mechanism could make the decoder's search for alignments more tractable and more scalable.

In this work, we perform planning over the input sequence by searching for alignments; our model does not form an explicit plan of the output tokens to generate. Nevertheless, we find this alignment-based planning to improve performance significantly in several tasks, including character-level NMT. Planning can also be applied explicitly to generation in sequence-to-sequence tasks. For example, recent work by Bahdanau et al. (2016) on actor-critic methods for sequence prediction can be seen as this kind of generative planning.

We evaluate our model and report results on character-level translation tasks from WMT'15 for English to German, English to Finnish, and English to Czech language pairs. On almost all pairs we observe improvements over a baseline that represents the state-of-the-art in neural character-level translation. In our NMT experiments, our model outperforms the baseline despite using significantly fewer parameters and converges faster in training. We also show that our model performs better than strong baselines on the algorithmic task of finding Eulerian circuits in random graphs and the task of natural-language question generation from a document and target answer.

## 2   Related Works

Existing sequence-to-sequence models with attention have focused on generating the target sequence by aligning each generated output token to another token in the input sequence. This approach has proven successful in neural machine translation (Bahdanau et al., 2016) and has recently been adapted to several other applications, including speech recognition (Chan et al., 2015) and image caption generation (Xu et al., 2015). In general these models construct alignments using a simple MLP that conditions on the decoder's internal state. In our work we integrate a planning mechanism into the alignment function.

There have been several earlier proposals for different alignment mechanisms: for instance, Yang et al. (2016) developed a hierarchical attention mechanism to perform document-level classification, while Luo et al. (2016) proposed an algorithm for learning discrete alignments between two sequences using policy gradients (Williams, 1992).

Silver et al. (2016) used a planning mechanism based on Monte Carlo tree search with neural networks to train reinforcement learning (RL) agents on the game of Go. Most similar to our work, Vezhnevets et al. (2016) developed a neural planning mechanism, called the strategic attentive reader and writer (STRAW), that can learn high-level temporally abstracted macro-actions. STRAW uses an action plan matrix, which represents the sequences of actions the model plans to take, and a commitment plan vector, which determines whether to commit an action or recompute the plan. STRAW's action plan and commitment plan are stochastic and the model is trained with RL. Our model computes an alignment plan rather than an action plan, and both its alignment matrix and commitment vector are deterministic and end-to-end trainable with backpropagation.

Our experiments focus on character-level neural machine translation because learning alignments for long sequences is difficult for existing models. This effect can be more pronounced in character-level NMT, since sequences of characters are longer than corresponding sequences of words. Furthermore, to learn a proper alignment between sequences a model often must learn to segment them correctly, a process suited to planning. Previously, Chung et al. (2016) and Lee et al. (2016) addressed the character-level machine translation problem with architectural modifications to the encoder and the decoder. Our model is the first we are aware of to tackle the problem through planning.

## 3 Planning for Sequence-to-Sequence Learning

We now describe how to integrate a planning mechanism into a sequence-to-sequence architecture with attention (Bahdanau et al., 2015). Our model first creates a *plan*, then computes a soft *alignment* based on the plan, and *generates* at each time-step in the decoder. We refer to our model as PAG (Plan-Attend-Generate).

### 3.1 Notation and Encoder

As input our model receives a sequence of tokens, $X = (x_0, \cdots, x_{|X|})$, where $|X|$ denotes the length of $X$. It processes these with the encoder, a bidirectional RNN. At each input position $i$ we obtain annotation vector $\mathbf{h}_i$ by concatenating the forward and backward encoder states, $\mathbf{h}_i = [\mathbf{h}_i^{\rightarrow}; \mathbf{h}_i^{\leftarrow}]$, where $\mathbf{h}_i^{\rightarrow}$ denotes the hidden state of the encoder's forward RNN and $\mathbf{h}_i^{\leftarrow}$ denotes the hidden state of the encoder's backward RNN.

Through the decoder the model predicts a sequence of output tokens, $Y = (y_1, \cdots, y_{|Y|})$. We denote by $\mathbf{s}_t$ the hidden state of the decoder RNN generating the target output token at time-step $t$.

### 3.2 Alignment and Decoder

Our goal is a mechanism that plans which parts of the input sequence to focus on for the next $k$ time-steps of decoding. For this purpose, our model computes an alignment plan matrix $\mathbf{A}_t \in \mathbb{R}^{k \times |X|}$ and commitment plan vector $\mathbf{c}_t \in \mathbb{R}^k$ at each time-step. Matrix $\mathbf{A}_t$ stores the alignments for the current and the next $k-1$ timesteps; it is conditioned on the current input, i.e. the token predicted at the previous time-step, $\mathbf{y}_t$, and the current context $\psi_t$, which is computed from the input annotations $\mathbf{h}_i$. Each row of $\mathbf{A}_t$ gives the logits for a probability vector over the input annotation vectors. The first row gives the logits for the current time-step, $t$, the second row for the next time-step, $t+1$, and so on. The recurrent decoder function, $f_{\text{dec-rnn}}(\cdot)$, receives $\mathbf{s}_{t-1}$, $\mathbf{y}_t$, $\psi_t$ as inputs and computes the hidden state vector

$$\mathbf{s}_t = f_{\text{dec-rnn}}(\mathbf{s}_{t-1}, \mathbf{y}_t, \psi_t). \tag{1}$$

Context $\psi_t$ is obtained by a weighted sum of the encoder annotations,

$$\psi_t = \sum_i^{|X|} \alpha_{ti} \mathbf{h}_i, \tag{2}$$

where the soft-alignment vector $\alpha_t = \texttt{softmax}(\mathbf{A}_t[0]) \in \mathbb{R}^{|X|}$ is a function of the first row of the alignment matrix. At each time-step, we compute a candidate alignment-plan matrix $\bar{\mathbf{A}}_t$ whose entry at the $i^{th}$ row is

$$\bar{\mathbf{A}}_t[i] = f_{\text{align}}(\mathbf{s}_{t-1}, \mathbf{h}_j, \beta_t^i, \mathbf{y}_t), \tag{3}$$

where $f_{\text{align}}(\cdot)$ is an MLP and $\beta_t^i$ denotes a summary of the alignment matrix's $i^{th}$ row at time $t-1$. The summary is computed using an MLP, $f_r(\cdot)$, operating row-wise on $\mathbf{A}_{t-1}$: $\beta_t^i = f_r(\mathbf{A}_{t-1}[i])$.

The commitment plan vector $\mathbf{c}_t$ governs whether to follow the existing alignment plan, by shifting it forward from $t-1$, or to recompute it. Thus, $\mathbf{c}_t$ represents a discrete decision. For the model to operate discretely, we use the recently proposed Gumbel-Softmax trick (Jang et al., 2016; Maddison et al., 2016) in conjunction with the straight-through estimator (Bengio et al., 2013) to backpropagate through $\mathbf{c}_t$.[1] The model further learns the temperature for the Gumbel-Softmax as proposed in (Gulcehre et al., 2017). Both the commitment vector and the action plan matrix are initialized with ones; this initialization is not modified through training.

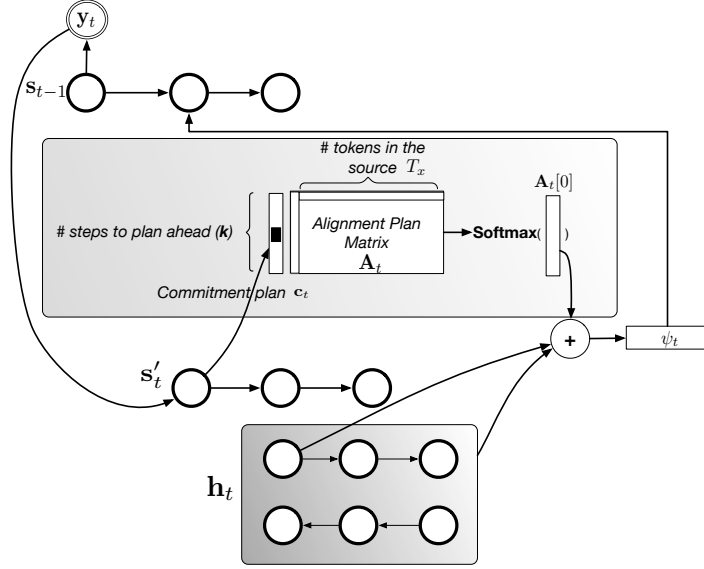

Figure 1: Our planning mechanism in a sequence-to-sequence model that learns to plan and execute alignments. Distinct from a standard sequence-to-sequence model with attention, rather than using a simple MLP to predict alignments our model makes a plan of future alignments using its alignment-plan matrix and decides when to follow the plan by learning a separate commitment vector. We illustrate the model for a decoder with two layers $\mathbf{s}_t'$ for the first layer and the $\mathbf{s}_t$ for the second layer of the decoder. The planning mechanism is conditioned on the first layer of the decoder ($\mathbf{s}_t'$).

**Alignment-plan update**  Our decoder updates its alignment plan as governed by the commitment plan. We denote by $g_t$ the first element of the discretized commitment plan $\bar{\mathbf{c}}_t$. In more detail, $g_t = \bar{\mathbf{c}}_t[0]$, where the discretized commitment plan is obtained by setting $\mathbf{c}_t$'s largest element to 1 and all other elements to 0. Thus, $g_t$ is a binary indicator variable; we refer to it as the commitment switch. When $g_t = 0$, the decoder simply advances the time index by shifting the action plan matrix $\mathbf{A}_{t-1}$ forward via the shift function $\rho(\cdot)$. When $g_t = 1$, the controller reads the action-plan matrix to produce the summary of the plan, $\beta_t^i$. We then compute the updated alignment plan by interpolating the previous alignment plan matrix $\mathbf{A}_{t-1}$ with the candidate alignment plan matrix $\bar{\mathbf{A}}_t$. The mixing ratio is determined by a learned update gate $\mathbf{u}_t \in \mathbb{R}^{k \times |X|}$, whose elements $\mathbf{u}_{ti}$ correspond to tokens in the input sequence and are computed by an MLP with sigmoid activation, $f_{\mathrm{up}}(\cdot)$:

$$\mathbf{u}_{ti} = f_{\mathrm{up}}(\mathbf{h}_i, \mathbf{s}_{t-1}),$$
$$\mathbf{A}_t[:,i] = (1 - \mathbf{u}_{ti}) \odot \mathbf{A}_{t-1}[:,i] + \mathbf{u}_{ti} \odot \bar{\mathbf{A}}_t[:,i].$$

To reiterate, the model only updates its alignment plan when the current commitment switch $g_t$ is active. Otherwise it uses the alignments planned and committed at previous time-steps.

**Commitment-plan update**  The commitment plan also updates when $g_t$ becomes 1. If $g_t$ is 0, the shift function $\rho(\cdot)$ shifts the commitment vector forward and appends a 0-element. If $g_t$ is 1, the model recomputes $\mathbf{c}_t$ using a single layer MLP, $f_c(\cdot)$, followed by a Gumbel-Softmax, and $\bar{\mathbf{c}}_t$ is recomputed by discretizing $\mathbf{c}_t$ as a one-hot vector:

$$\mathbf{c}_t = \texttt{gumbel\_softmax}(f_c(\mathbf{s}_{t-1})), \qquad (4)$$
$$\bar{\mathbf{c}}_t = \texttt{one\_hot}(\mathbf{c}_t). \qquad (5)$$

We provide pseudocode for the algorithm to compute the commitment plan vector and the action plan matrix in Algorithm 1. An overview of the model is depicted in Figure 1.

### 3.2.1 Alignment Repeat

In order to reduce the model's computational cost, we also propose an alternative to computing the candidate alignment-plan matrix at every step. Specifically, we propose a model variant that reuses the

**Algorithm 1:** Pseudocode for updating the alignment plan and commitment vector.

---

**for** $j \in \{1, \cdots |X|\}$ **do**
    **for** $t \in \{1, \cdots |Y|\}$ **do**
        **if** $g_t = 1$ **then**
            $\mathbf{c}_t = \mathrm{softmax}(f_c(\mathbf{s}_{-1}))$
            $\beta_t^j = f_r(\mathbf{A}_{t-1}[j])$ {Read alignment plan}
            $\bar{\mathbf{A}}_t[i] = f_{\mathrm{align}}(\mathbf{s}_{t-1}, \mathbf{h}_j, \beta_t^j, \mathbf{y}_t)$ {Compute candidate alignment plan}
            $\mathbf{u}_{tj} = f_{\mathrm{up}}(\mathbf{h}_j, \mathbf{s}_{t-1}, \psi_{t-1})$ {Compute update gate}
            $\mathbf{A}_t = (1 - \mathbf{u}_{tj}) \odot \mathbf{A}_{t-1} + \mathbf{u}_{tj} \odot \bar{\mathbf{A}}_t$ {Update alignment plan}
        **else**
            $\mathbf{A}_t = \rho(\mathbf{A}_{t-1})$ {Shift alignment plan}
            $\mathbf{c}_t = \rho(\mathbf{c}_{t-1})$ {Shift commitment plan}
        **end if**
        Compute the alignment as $\alpha_t = \mathrm{softmax}(\mathbf{A}_t[0])$
    **end for**
**end for**

---

alignment *vector* from the previous time-step until the commitment switch activates, at which time the model computes a new alignment vector. We call this variant *repeat, plan, attend, and generate* (rPAG). rPAG can be viewed as learning an explicit segmentation with an implicit planning mechanism in an unsupervised fashion. Repetition can reduce the computational complexity of the alignment mechanism drastically; it also eliminates the need for an explicit alignment-plan matrix, which reduces the model's memory consumption also. We provide pseudocode for rPAG in Algorithm 2.

**Algorithm 2:** Pseudocode for updating the repeat alignment and commitment vector.

---

**for** $j \in \{1, \cdots |X|\}$ **do**
    **for** $t \in \{1, \cdots |Y|\}$ **do**
        **if** $g_t = 1$ **then**
            $\mathbf{c}_t = \mathrm{softmax}(f_c(\mathbf{s}_{t-1}, \psi_{t-1}))$
            $\alpha_t = \mathrm{softmax}(f_{\mathrm{align}}(\mathbf{s}_{t-1}, \mathbf{h}_j, \mathbf{y}_t))$
        **else**
            $\mathbf{c}_t = \rho(\mathbf{c}_{t-1})$ {Shift the commitment vector $\mathbf{c}_{t-1}$}
            $\alpha_t = \alpha_{t-1}$ {Reuse the old the alignment}
        **end if**
    **end for**
**end for**

---

### 3.3  Training

We use a deep output layer (Pascanu et al., 2013a) to compute the conditional distribution over output tokens,

$$p(\mathbf{y}_t | \mathbf{y}_{<t}, \mathbf{x}) \propto \mathbf{y}_t^\top \exp(\mathbf{W}_o f_o(\mathbf{s}_t, \mathbf{y}_{t-1}, \psi_t)), \tag{6}$$

where $\mathbf{W}_o$ is a matrix of learned parameters and we have omitted the bias for brevity. Function $f_o$ is an MLP with $\tanh$ activation.

The full model, including both the encoder and decoder, is jointly trained to minimize the (conditional) negative log-likelihood

$$\mathcal{L} = -\frac{1}{N} \sum_{n=1}^{N} \log p_\theta(\mathbf{y}^{(n)} | \mathbf{x}^{(n)}),$$

where the training corpus is a set of $(\mathbf{x}^{(n)}, \mathbf{y}^{(n)})$ pairs and $\boldsymbol{\theta}$ denotes the set of all tunable parameters. As noted by Vezhnevets et al. (2016), the proposed model can learn to recompute very often, which decreases the utility of planning. To prevent this behavior, we introduce a loss that penalizes the model for committing too often,

$$\mathcal{L}_{\mathrm{com}} = \lambda_{\mathrm{com}} \sum_{t=1}^{|X|} \sum_{i=0}^{k} || \frac{1}{k} - \mathbf{c}_{ti} ||_2^2, \tag{7}$$

where $\lambda_{\mathrm{com}}$ is the commitment hyperparameter and $k$ is the timescale over which plans operate.

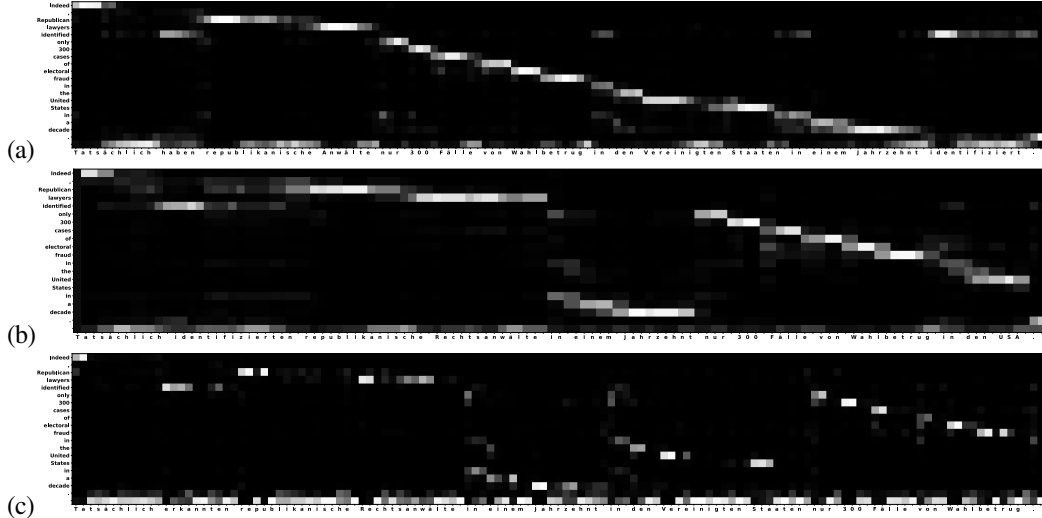

Figure 2: We visualize the alignments learned by PAG in (a), rPAG in (b), and our baseline model with a 2-layer GRU decoder using $\mathbf{h}_2$ for the attention in (c). As depicted, the alignments learned by PAG and rPAG are smoother than those of the baseline. The baseline tends to put too much attention on the last token of the sequence, defaulting to this empty location in alternation with more relevant locations. Our model, however, places higher weight on the last token usually when no other good alignments exist. We observe that rPAG tends to generate less monotonic alignments in general.

## 4 Experiments

Our baseline is the encoder-decoder architecture with attention described in Chung et al. (2016), wherein the MLP that constructs alignments conditions on the second-layer hidden states, $\mathbf{h}^2$, in the two-layer decoder. The integration of our planning mechanism is analogous across the family of attentive encoder-decoder models, thus our approach can be applied more generally. In all experiments below, we use the same architecture for our baseline and the (r)PAG models. The only factor of variation is the planning mechanism. For training all models we use the Adam optimizer with initial learning rate set to $0.0002$. We clip gradients with a threshold of $5$ (Pascanu et al., 2013b) and set the number of planning steps ($k$) to 10 throughout. In order to backpropagate through the alignment-plan matrices and the commitment vectors, the model must maintain these in memory, increasing the computational overhead of the PAG model. However, rPAG does not suffer from these computational issues.

### 4.1 Algorithmic Task

We first compared our models on the algorithmic task from Li et al. (2015) of finding the "Eulerian Circuits" in a random graph. The original work used random graphs with 4 nodes only, but we found that both our baseline and the PAG model solve this task very easily. We therefore increased the number of nodes to 7. We tested the baseline described above with hidden-state dimension of 360, and the same model augmented with our planning mechanism. The PAG model solves the Eulerian Circuits problem with 100% absolute accuracy on the test set, indicating that for all test-set graphs, all nodes of the circuit were predicted correctly. The baseline encoder-decoder architecture with attention performs well but significantly worse, achieving 90.4% accuracy on the test set.

### 4.2 Question Generation

SQUAD (Rajpurkar et al., 2016) is a question answering (QA) corpus wherein each sample is a (document, question, answer) triple. The document and the question are given in words and the answer is a span of word positions in the document. We evaluate our planning models on the recently proposed question-generation task (Yuan et al., 2017), where the goal is to generate a question conditioned on a document and an answer. We add the planning mechanism to the encoder-decoder architecture proposed by Yuan et al. (2017). Both the document and the answer are encoded via recurrent neural networks, and

the model learns to align the question output with the document during decoding. The pointer-softmax mechanism (Gulcehre et al., 2016) is used to generate question words from either a shortlist vocabulary or by copying from the document. Pointer-softmax uses the alignments to predict the location of the word to copy; thus, the planning mechanism has a direct influence on the decoder's predictions.

We used 2000 examples from SQUAD's training set for validation and used the official development set as a test set to evaluate our models. We trained a model with 800 units for all GRU hidden states 600 units for word embedding. On the test set the baseline achieved 66.25 NLL while PAG got 65.45 NLL. We show the validation-set learning curves of both models in Figure 3.

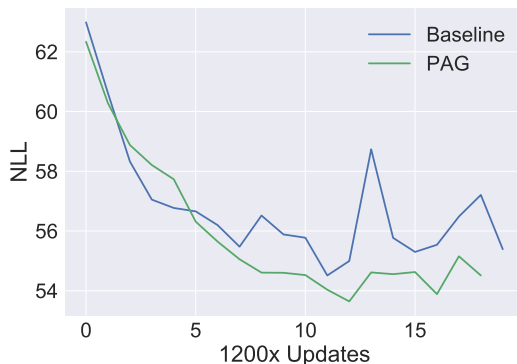

Figure 3: Learning curves for question-generation models on our development set. Both models have the same capacity and are trained with the same hyperparameters. PAG converges faster than the baseline with better stability.

## 4.3 Character-level Neural Machine Translation

Character-level neural machine translation (NMT) is an attractive research problem (Lee et al., 2016; Chung et al., 2016; Luong and Manning, 2016) because it addresses important issues encountered in word-level NMT. Word-level NMT systems can suffer from problems with rare words (Gulcehre et al., 2016) or data sparsity, and the existence of compound words without explicit segmentation in some language pairs can make learning alignments between different languages and translations more difficult. Character-level neural machine translation mitigates these issues.

In our NMT experiments we use byte pair encoding (BPE) (Sennrich et al., 2015) for the source sequence and characters at the target, the same setup described in Chung et al. (2016). We also use the same preprocessing as in that work.[2] We present our experimental results in Table 1. Models were tested on the WMT'15 tasks for English to German (En→De), English to Czech (En→Cs), and English to Finnish (En→Fi) language pairs. The table shows that our planning mechanism improves translation performance over our baseline (which reproduces the results reported in (Chung et al., 2016) to within a small margin). It does this with fewer updates and fewer parameters. We trained (r)PAG for 350K updates on the training set, while the baseline was trained for 680K updates. We used 600 units in (r)PAG's encoder and decoder, while the baseline used 512 in the encoder and 1024 units in the decoder. In total our model has about 4M fewer parameters than the baseline. We tested all models with a beam size of 15.

As can be seen from Table 1, layer normalization (Ba et al., 2016) improves the performance of PAG significantly. However, according to our results on En→De, layer norm affects the performance of rPAG only marginally. Thus, we decided not to train rPAG with layer norm on other language pairs.

In Figure 2, we show qualitatively that our model constructs smoother alignments. At each word that the baseline decoder generates, it aligns the first few characters to a word in the source sequence, but for the remaining characters places the largest alignment weight on the last, empty token of the source sequence. This is because the baseline becomes confident of which word to generate after the first few characters, and it generates the remainder of the word mainly by relying on language-model predictions. We observe that (r)PAG converges faster with the help of the improved alignments, as illustrated by the learning curves in Figure 4.

| | Model | Layer Norm | Dev | Test 2014 | Test 2015 |
|---|---|---|---|---|---|
| En→De | Baseline | ✗ | 21.57 | 21.33 | **23.45** |
| | Baseline† | ✗ | 21.4 | 21.16 | 22.1 |
| | Baseline† | ✓ | 21.65 | 21.69 | 22.55 |
| | PAG | ✗ | 21.92 | 21.93 | 22.42 |
| | | ✓ | **22.44** | **22.59** | **23.18** |
| | rPAG | ✗ | 21.98 | 22.17 | 22.85 |
| | | ✓ | 22.33 | 22.35 | 22.83 |
| En→Cs | Baseline | ✗ | 17.68 | 19.27 | 16.98 |
| | Baseline† | ✓ | 19.1 | 21.35 | 18.79 |
| | PAG | ✗ | 18.9 | 20.6 | 18.88 |
| | | ✓ | **19.44** | **21.64** | **19.48** |
| | rPAG | ✗ | 18.66 | 21.18 | 19.14 |
| En→Fi | Baseline | ✗ | 11.19 | - | 10.93 |
| | Baseline† | ✓ | 11.26 | - | 10.71 |
| | PAG | ✗ | 12.09 | - | 11.08 |
| | | ✓ | **12.85** | - | **12.15** |
| | rPAG | ✗ | 11.76 | - | 11.02 |

Table 1: The results of different models on the WMT'15 tasks for English to German, English to Czech, and English to Finnish language pairs. We report BLEU scores of each model computed via the *multi-blue.perl* script. The best-score of each model for each language pair appears in bold-face. We use *newstest2013* as our development set, *newstest2014* as our "Test 2014" and *newstest2015* as our "Test 2015" set. ($^{\dagger}$) denotes the results of the baseline that we trained using the hyperparameters reported in Chung et al. (2016) and the code provided with that paper. For our baseline, we only report the median result, and do not have multiple runs of our models. On WMT'14 and WMT'15 for En$rightarrow$De character-level NMT, Kalchbrenner et al. (2016) have reported better results with deeper auto-regressive convolutional models (Bytenets), 23.75 and 26.26 respectively.

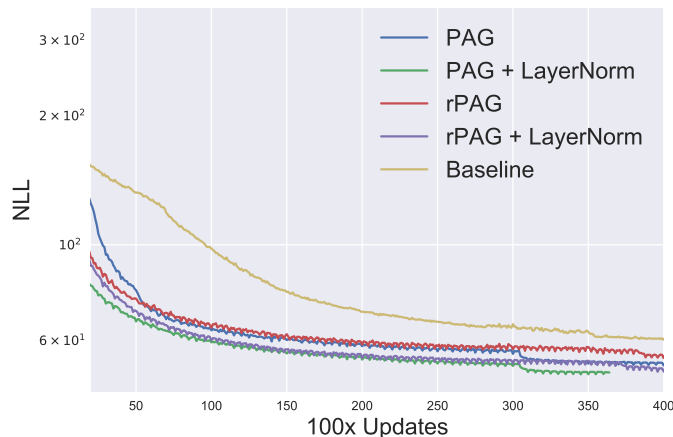

Figure 4: Learning curves for different models on WMT'15 for En→De. Models with the planning mechanism converge faster than our baseline (which has larger capacity).

## 5 Conclusion

In this work we addressed a fundamental issue in neural generation of long sequences by integrating *planning* into the alignment mechanism of sequence-to-sequence architectures. We proposed two different planning mechanisms: PAG, which constructs explicit plans in the form of stored matrices, and rPAG, which plans implicitly and is computationally cheaper. The (r)PAG approach empirically improves alignments over long input sequences. We demonstrated our models' capabilities through results on

character-level machine translation, an algorithmic task, and question generation. In machine translation, models with planning outperform a state-of-the-art baseline on almost all language pairs using fewer parameters. We also showed that our model outperforms baselines with the same architecture (minus planning) on question-generation and algorithmic tasks. The introduction of planning improves training convergence and potentially the speed by using the alignment repeats.

## Footnotes

[1] We also experimented with training $\mathbf{c}_t$ using REINFORCE (Williams, 1992) but found that Gumbel-Softmax led to better performance.

[2]Our implementation is based on the code available at `https://github.com/nyu-dl/dl4mt-cdec`

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
