[Reviews · NeurIPS 2017]

Reviewer 1



The authors have addressed the problem of sequence-to-sequence modelling. Specifically, they investigate a mechanism which allows planning future alignments while computing current alignment of input and output sequences. The proposed method outperforms the baseline methods in three different tasks with faster convergence and fewer parameters. Pros: * The paper is very well written and easy to follow. * The idea of using a planning mechanism on sequence-to-sequence models seems novel. * The proposed method is able to outperform baseline methods on different tasks which suggests that it can be a good fit for general sequence-to-sequence modeling tasks. * Also, the proposed method requires less parameters and shows faster and better convergence. Cons: * It would have been nicer to see the effect of number of planning steps on the performance. The choice of 10 planning steps is not justified.

Reviewer 2



The paper introduces the planning mechanism into the encoder-decoder model with attention. The key idea is that instead of computing the attention weights on-the-fly for each decoding time step, an attention template is generated for the next k time steps based on the current decoding hidden state s_{t-1}, and a commitment vector is used to decide whether to follow this alignment plan or to recompute it. Overall, the paper is well written, and technical discussion are clear. The effect of the planning mechanism is demonstrated by Figure 2, where planning mechanism tends to generate smoother alignments compared to the vanilla attention approach. In this paper, the attention template (matrix A) is computed as eq(3). It would be interesting to see if some other type of features could be used, and if they could control the language generation, and produce more diverse and controllable responses. This might be evaluated on a conversation task, however, I understand that it is out of the scope of this paper. In general, this is an interesting idea to me. A couple of minor questions: -- In Table 1, why the Baseline + layer norm is missing? PAG and rPAG seem only outperform the baseline when layer norm is used. --You used k=10 for all the experiments. Is there any motivation behind that? 10 time steps only correspond to 1 to 2 English words for character MT. --Do you have any results with word-level MT? Does the planning approach help? --Can you say anything about the computation complexity? Does it slow down training a lot?

Reviewer 3



This paper addresses the problem of using neural models to generate long sequences. In particular, the authors seek to modify the decoder of the encoder-decoder architecture as used in sequence to sequence models. The encoder is fixed, as is used as in prior work What the authors propose is a plan-ahead alignment mechanism. A commitment vector is used to decide wether to update the commitment plan or to follow existing the plan ahead alignment. Experiments were carried out on a random graphs task, a question generation task, and character level machine translation. 1. In terms of the contribution of the paper, a lookahead alignment mechanism, while interesting, and well motivated in the paper, is a bit unclear if it is significant enough a contribution to carry the paper. 2. The experimental evaluation also does not given one enough confidence. First, the first two experiments are not very insightful, in particular, for the experiment on question generation, the main result given is a convergence plot. The third experiment, on NMT is equally minimal, and has only a very small comparison to prior work. 3. Overall, the paper’s contribution is somewhat limited, and one wishes it was better evaluated, the current experiments are somewhat minimal.